# Permeability of the Retina and RPE-Choroid-Sclera to Three Ophthalmic Drugs and the Associated Factors

**DOI:** 10.3390/pharmaceutics13050655

**Published:** 2021-05-04

**Authors:** Hyeong Min Kim, Hyounkoo Han, Hye Kyoung Hong, Ji Hyun Park, Kyu Hyung Park, Hyuncheol Kim, Se Joon Woo

**Affiliations:** 1Department of Ophthalmology, Seoul National University College of Medicine, Seoul National University Bundang Hospital, Seongnam 13620, Korea; hmkim3@gmail.com (H.M.K.); alpaomega@hanmail.net (H.K.H.); park9845@hanmail.net (J.H.P.); jiani4@snu.ac.kr (K.H.P.); 2Department of Chemical and Biomolecular Engineering, Sogang University, Seoul 04107, Korea; 507513@hanmail.net

**Keywords:** permeability, retina, retinal pigment epithelium, Ussing chamber, intravitreal half-life

## Abstract

In this study, Retina-RPE-Choroid-Sclera (RCS) and RPE-Choroid-Sclera (CS) were prepared by scraping them off neural retina, and using the Ussing chamber we measured the average time–concentration values in the acceptor chamber across five isolated rabbit tissues for each drug molecule. We determined the outward direction permeability of the RCS and CS and calculated the neural retina permeability. The permeability coefficients of RCS and CS were as follows: ganciclovir, 13.78 ± 5.82 and 23.22 ± 9.74; brimonidine, 15.34 ± 7.64 and 31.56 ± 12.46; bevacizumab, 0.0136 ± 0.0059 and 0.0612 ± 0.0264 (×10^−6^ cm/s). The calculated permeability coefficients of the neural retina were as follows: ganciclovir, 33.89 ± 12.64; brimonidine, 29.83 ± 11.58; bevacizumab, 0.0205 ± 0.0074 (×10^−6^ cm/s). Between brimonidine and ganciclovir, lipophilic brimonidine presented better RCS and CS permeability, whereas ganciclovir showed better calculated neural retinal permeability. The large molecular weight drug bevacizumab demonstrated a much lower permeability than brimonidine and ganciclovir. In conclusion, the ophthalmic drug permeability of RCS and CS is affected by the molecular weight and lipophilicity, and influences the intravitreal half-life.

## 1. Introduction

The advancement of ophthalmic drug delivery to the posterior segment requires extensive knowledge of intraocular pharmacokinetics (PK). Representative intraocular PK parameters, such as intravitreal half-life, mean residence time, observed maximum concentration, and clearance rate, were considered as the substantial indicators of drug efficacy [1,2]. Thus, numerous investigations have been conducted to establish better drug efficacy with intravitreal half-life prolongation by reducing the elimination rate and controlling drug release [3,4,5]. Regarding the elimination/clearance process, retinal pigment epithelium (RPE) is considered as a major barrier in the posterior segment ocular delivery, which is located between the neural retina and choroid [1,6]. After intravitreal injection of newly developed anti-vascular endothelial growth factor (VEGF) drugs or small molecular drugs, the molecules diffuse in the intraocular space. These are eliminated via the anterior (aqueous humor) and posterior (retina-RPE-choroid) pathways across the blood-ocular barriers, including the blood-aqueous barrier (BAB) and blood-retinal barrier (BRB), which consists of the retinal pigment epithelium (RPE) and endothelium of the retinal vessels [7,8,9]. In addition, the inner limiting membrane (ILM), which is composed of a fine meshwork with nanopores, and tight junctions in the RPE, are considered to restrict drug permeation [10,11]. Therefore, drug permeability in the posterior segment, specifically the neural retina, RPE, and choroid, is important to achieve better drug efficacy.

Since retinal permeability is crucial for intraocular pharmacokinetics and dynamics, numerous experiments have been conducted previously, which found that RPE drug permeability was largely affected by molecular physicochemical properties such as molecular weight (MW) and lipophilicity [12,13,14]. These permeability studies were performed with ex vivo ocular tissues using a special diffusion apparatus known as the “Ussing chamber”, which is widely adopted for the evaluation of molecular transport across epithelia [15]. The ionic transport mechanisms in the fresh isolated tissues were identified and the retinal permeability was estimated by measuring the drug concentration in the chamber and recording the transepithelial resistance/potential. However, most of the previous investigations have focused on the trans-scleral delivery of ophthalmic drugs, from beta-blockers to immunoglobulin and fluorescein isothiocyanate (FITC)-labeled dextrans [12,16,17]. Currently, a few permeability studies have used intravitreal injection drugs [18,19], but experimental data are insufficient to keep up with the rapid advancement of posterior segment ocular drug delivery.

Thus, the purpose of our study was to provide systematic data on the pre-clinically measured permeability coefficients in posterior segment tissues, and to help clinicians and researchers to assess the intraocular pharmacokinetics of three substantial drug compounds being investigated and engineered for the advancement of ocular drug delivery systems. Prior investigations concentrating on the retinal permeability and its effect on intraocular PK parameters have not been adequate, especially the measurement or calculation of the experimental values. Therefore, in this study, we experimented with three ophthalmic drugs, including ganciclovir, brimonidine, and bevacizumab, to evaluate retinal and RPE-choroid-sclera permeability from the vitreous to the systemic blood circulation (outward direction) in New Zealand white rabbit eyes. As mentioned above, both ganciclovir and brimonidine are small molecular weight drugs, and sustained-release intravitreal implants have been developed. Bevacizumab is a representative large molecular weight anti-VEGF drug that has been applied to various retinal diseases. Then, together with seven different small drug molecules studied in previous literature, we analyzed the association between intravitreal half-life and retinal permeability in various ophthalmic drugs.

## 2. Materials and Methods

### 2.1. Isolation and Preparation of Ocular Tissues

Approval was obtained from the Seoul National University Bundang Hospital Institutional Animal Care and Use Committee, and animal experiments were conducted in accordance with the guidelines of the Association for Research in Vision and Ophthalmology for animal use in research.

A total of 30 eyes from 15 healthy New Zealand white rabbits weighing 1.5–2.0 kg and aged 8–10 weeks were used for the study; five eyes each for the three ophthalmic drugs (brimonidine, ganciclovir, and bevacizumab) with two different ocular tissues (retina-RPE-choroid-sclera and RPE-choroid-sclera). The rabbits were anesthetized with 15 mg/kg tiletamine hydrochloride/zolazepam hydrochloride (Zoletil, Virbac Laboratories, Carros Cedex, France) and 5 mg/kg of xylazine hydrochloride (Rompun, Bayer Healthcare, Seoul, Korea), and the rabbit eyes were enucleated immediately after euthanization and stored in ice-cold Ringer buffer until use. We then removed the adherent muscles and fats on the outer sclera, divided the anterior and posterior segments by circumferential cutting behind the limbus, and gently separated the lens and vitreous from the neural retina to obtain retina-RPE-choroid-sclera tissues. Finally, we carefully detached the neural retina using forceps to obtain RPE-choroid-sclera tissues. Isolated tissues were stored in phosphate-buffered saline (PBS; pH 7.4). To minimize the damage to tissue integrity, the extracted tissues were used immediately.

### 2.2. Permeability Studies with the Ussing Chamber

Two different ocular tissues (retina-RPE-choroid-sclera and RPE-choroid-sclera) were placed in a tissue insert with ring-shaped silicone adapters and a circular diffusion area of 0.19625 cm^2^ which was enclosed with micropore membranes (0.22 μm, nitrocellulose; MILLIPORE, Darmstadt, Germany), and mounted in the Ussing chamber (Navicyte, Harvard Apparatus, Holliston, MA, USA). Both sides of the chamber (left, donor chamber; right, acceptor chamber) were filled with 4 mL of PBS solution, with drug concentrations of ganciclovir and brimonidine (Sigma-Aldrich, St. Louis, MO, USA) dissolved at 0.244 mg/mL, and bevacizumab (Avastin; Genentech, Inc., San Francisco, CA, USA) at 4.469 mg/mL on the donor side. Both compartments in the Ussing chamber were filled with oxygen or a mixture of 95% O_2_ and 5% CO_2_. Permeability was measured only in the outward direction, which resembles the elimination pathway from the vitreous to the choroid to the systemic blood circulation.

Samples (0.1 mL) were taken from the acceptor chambers at eight different time points (0.5, 1, 1.5, 2, 2.5, 3, 3.5, and 4 h) and replaced with fresh buffer of the same volume. For the small molecular weight drugs, ganciclovir and brimonidine, drug concentrations from acceptor chambers were quantified by high-performance liquid chromatography (YL9100 HPLC system, Youngin, Korea). Zorbax Eclipse Plus C18 (4.6 mm × 250 mm, 5 μm; Agilent Technologies, Santa Clara, CA, USA) was used for the separation of brimonidine. The flow rate was 1.0 mL/min, and elution of the mobile phase of acetonitrile:buffer (10:90, *v*/*v*) was isocratic. The buffer consisted of 10 mM trimethylamine, adjusted to pH 3.2 with phosphoric acid. The UV detector was set to 248 nm, and the retention time was 6 min. Symmetry C18 (4.6 × 250 mm, 5 μm; Waters, Milford, MA, USA) was used for the separation of ganciclovir. The flow rate was 1.0 mL/min, and the mobile phase of methanol:buffer (20:80, *v*/*v*) was isocratic. The buffer was prepared by diluting 1 mL of trifluoroacetic acid in 1000 mL distilled water, and the pH was adjusted to 2.5.

For the large molecular weight drug bevacizumab, an indirect enzyme-linked immunosorbent assay (ELISA) was performed to measure the drug concentrations. The specific immunoassay procedure was previously described in multiple articles published by our group [20,21,22,23,24]. In brief, the human 165-amino-acid variant of recombinant VEGF (R&D systems, Minneapolis, MN, USA) was diluted to 1.0 μg/mL in 50 mM carbonate buffer (pH 9), immobilized on 96-well flat-bottom plates (NUNC, Roskilde, Denmark), and aliquoted at 100 μL/well. Then, the plates were incubated overnight at 4 °C, washed with 1× PBS, and blocked for 2 h at 4 °C with 1% BSA in 1X PBS. After the final washing process, the plates were stored at 4 °C until further use. Subsequently, the samples acquired from the acceptor chambers at different time points were diluted with 0.1% BSA in 1X PBS to be within the range of the assay, aliquoted into the rVEGF-coated plates (100 μL/well), and incubated overnight at 4 °C. In each individual plate, known bevacizumab concentrations ranging from 0.049–3.125 ng/mL were included to generate a standard curve. Bevacizumab was detected using an anti-human immunoglobulin G horseradish peroxidase antibody (GE Healthcare, Pittsburgh, PA, USA). Finally, the optical density was measured by detecting the absorbance after treating the 3,3′,5,5′-tetramethyl benzidine substrate with hydrogen peroxide. Standard curves were generated using SoftMax Pro 5.4.1. software (Molecular Devices, Sunnyvale, CA, USA).

### 2.3. Selection of Ophthalmic Drugs for Permeability Experiments

In this study, three ophthalmic drugs, including ganciclovir, brimonidine, and bevacizumab, were selected for permeability measurements. Brimonidine is an α2-adrenergic agonist that is widely used in glaucoma treatment. Brimonidine is currently highlighted for its neuroprotective benefit by promoting retinal ganglion cell survival [25]. Moreover, as previously mentioned, a brimonidine intravitreal implant (Brimo DDS^®^, Allergan plc, Dublin, Ireland) is currently under human clinical trial and showed positive results in improving geographic atrophy in neovascular AMD eyes [26]. Thus, we decided to measure the posterior segment tissue permeability of brimonidine, since the drug might be used in the near future not only as a topical agent, but also for intravitreal administration. Ganciclovir is an antiviral drug for the treatment of cytomegalovirus (CMV) retinitis via an intravitreal injection [27]. Ganciclovir attracted our attention because we could assess the association of drug permeation and molecular physicochemical properties between brimonidine and ganciclovir, also considering that ganciclovir itself features a similar molecular weight, but is hydrophilic compared to brimonidine. In addition, ganciclovir has also been developed as an intravitreal implant (Vitrasert^®^, Bausch & Lomb, Rochester, NY, USA). Lastly, we selected bevacizumab to determine the effects of a large molecular weight on permeation. Identifying the permeability of these three ophthalmic drugs is crucial for posterior segment drug delivery to treat various retinal diseases.

### 2.4. Determination of Permeability Coefficients

The apparent permeability coefficients (*P*_app_) of the three ophthalmic drug molecules (ganciclovir, brimonidine, and bevacizumab) were calculated from the slope of the compound amount versus time profile on the acceptor side.
(1)$$P{\mathrm{app}}_{\;}=\left(\frac{\mathrm{dC}}{\mathrm{dt}}\right)\times V/\left(A\times C_{0}\right)$$

$\frac{\mathrm{dC}}{\mathrm{dt}}$: Change in the acceptor concentration calculated from the slope of the time–concentration curve between two time points

V: Buffer volume in the donor compartment (4 mL)

A: Diffusional area of the membrane (exposed surface area: 0.19625 cm^2^)

$C_{0}$: Initial donor chamber concentration

The slope of the time–concentration curve was derived from the concentration differences between the final time point (4 h) and the initial time point at the presumed steady-state, in this case, 30 min for brimonidine and ganciclovir, and 1.5 h for bevacizumab.

Since we conducted permeability studies with two different ocular tissues (retina-RPE-choroid-sclera (RCS) and RPE-choroid-sclera (CS) scraped off the neural retina), we calculated the neural retina permeability coefficients according to the equation below, which presents the overall contribution to the total permeability with multiple barriers and various permeation resistances.
(2)$$\frac{1}{P_{\mathrm{RCS}}}=\frac{1}{P_{R}}+\frac{1}{P_{\mathrm{CS}}}$$

## 3. Results

The average time–concentration values (μg/mL) of retina-RPE-choroid-sclera (RCS) and RPE-choroid-sclera (CS) rabbit ocular tissues in the acceptor chamber are shown in Table 1. For each ophthalmic drug in two separate tissues, five rabbit eyes (total 30 eyes) were sacrificed to measure the concentrations at eight different time points, from 30 min to 4 h. Then, we depicted the average time–concentration curves as in Figure 1, and calculated $\frac{\mathrm{dC}}{\mathrm{dt}}$ to derive the permeability coefficients using the equation. Regarding the linear fit in the time point versus average drug concentration plots, a correlation coefficient (R^2^) of over 0.9 was obtained. For the retina-RPE-choroid-sclera tissues, the linear fit model of ganciclovir was y = 0.3272x − 0.175 (R^2^ = 0.9613), brimonidine y = 0.3037x + 0.1143 (R^2^ = 0.9786), and bevacizumab y = 0.0087x − 0.0256 (R^2^ = 0.9237). For the RPE-choroid-sclera tissues, the linear fit model of ganciclovir was y = 0.8229x − 0.1262 (R^2^ = 0.9907), brimonidine y = 0.7948x + 0.8319 (R^2^ = 0.9965), and bevacizumab y = 0.0484x − 0.0703 (R^2^ = 0.9473). In this way, to derive every permeability coefficient for three ophthalmic drugs with two different ocular tissues, we calculated each $\frac{\mathrm{dC}}{\mathrm{dt}}$ value from the steady-state of the linear fit model in 30 rabbit eyes.

The representative physicochemical properties, molecular weight, lipophilicity (log P and water solubility), and the measured permeability coefficients from the experimental and calculated data (Table 1 and Figure 1) are summarized in Table 2. The permeability data are presented as the mean ± standard deviation. The permeability coefficients (*P*_app_) (×10^−6^ cm/s) of ganciclovir in retina-RPE-choroid-sclera and RPE-choroid-sclera were 13.78 ± 5.82 and 23.22 ± 9.74, brimonidine 15.34 ± 7.64 and 31.56 ± 12.46, and bevacizumab 0.0136 ± 0.0059 and 0.0612 ± 0.0264, respectively. Between the two small molecular weight drugs, brimonidine, which is more lipophilic with a higher log P and lower water solubility than ganciclovir, showed better permeability in RCS and CS tissues. On the other hand, bevacizumab, a much larger molecular weight drug compared to both ganciclovir and brimonidine, presented much worse permeability in RCS and CS tissues. According to the experimental data, we could estimate the permeability coefficients of the neural retina: ganciclovir 33.89 ± 12.64, brimonidine 29.83 ± 11.58, and bevacizumab 0.0205 ± 0.0074 (×10^−6^ cm/s). Unlike the RCS and CS, neural retinal permeability is greater in ganciclovir than in brimonidine, suggesting that rather than lipophilicity, the molecular weight might be the main pharmacokinetic (PK) parameter that affects drug molecule transport in the neural retinal layer.

In addition, using the experimental data of three ophthalmic drugs from this study, we evaluated the association between intravitreal half-life and posterior segment ocular tissue permeability. The log scale graph is shown in Figure 2, and we propose that the large molecule of bevacizumab with a longer intravitreal half-life, showed a lower permeability coefficient than brimonidine and ganciclovir. Experimental data with different species in the previous literature are presented in Table 3.

## 4. Discussion

In this study, we measured the retina-RPE-choroid-sclera and RPE-choroid-sclera permeability coefficients of three ophthalmic drugs, including ganciclovir, brimonidine, and bevacizumab, from rabbit ocular tissues, and calculated the neural retinal permeability coefficients. As a result, the large molecular weight biologic drug bevacizumab showed very restricted permeation across the posterior segment compared to small molecular weight drugs. Between ganciclovir and brimonidine, a more lipophilic drug, brimonidine, demonstrated better permeation through the RPE-choroid-sclera, even though its molecular weight is slightly larger than that of ganciclovir. However, ganciclovir was estimated to have better neural retinal permeability than brimonidine. Therefore, we propose that molecular weight might be the main factor affecting permeability in the neural retina, whereas both molecular weight and lipophilicity influence permeation through the retinal pigment epithelium (RPE) barrier. In addition, we could postulate that a longer intravitreal half-life might be negatively associated with posterior segment permeability, especially in the neural retina (Figure 2).

The retinal pigment epithelium (RPE) layer separates the neural retina from the choroid and exhibits intercellular tight junctions, working as a part of the blood-retinal barrier and restricting molecular transport. Therefore, numerous studies have focused on the mechanism of drug delivery across the RPE layer. To determine the RPE function for transport and permeation, the Ussing chamber was introduced as two separate baths divided by the epithelium, and electric potentials and concentrations were recorded to investigate the transepithelial ion transport [15]. Kimural et al. experimented in rabbit tissues with the Ussing chamber and suggested that the outward carboxyfluorescein movement through the RPE-choroid showed both passive and carrier-mediated active transport, despite the inward movement by passive diffusion [16]. Xhang et al. investigated brimonidine transport in RPE in bovine ocular tissues with the Ussing chamber, and discovered the temperature- and energy-dependent uptake of the drug molecules, which might be considered as evidence of a carrier-mediated transport process [38]. Skarphedinsdottir et al. examined the ion transport of mouse retina-choroid-sclera tissues with the Ussing chamber and concluded that Na–K adenosine triphosphatase (ATP) and Na–K-2Cl cotransporters are accountable for transepithelial ion transport [39].

Together with the RPE function of molecular transport, considerable efforts have been made to determine the relationship between molecular physicochemical properties and diffusion/permeation in the posterior segment of ocular tissues. Ambati et al. investigated the in vitro scleral permeability of high molecular weight compounds in rabbit eyes with the Ussing chamber, and proposed that the scleral permeability was negatively correlated with molecular weight and radius [12]. Pitkanen et al. experimented with RPE-choroid bovine eye tissues and found that the molecular weight and lipophilicity were relevant to the molecular permeability. The authors suggested that the RPE layer might be a major barrier in the posterior segment drug delivery through the trans-scleral route [13]. Kadam et al. evaluated RPE-choroid-sclera (CS) transport of eight β-blockers in human and various animal ocular tissues. According to this study, RPE-choroid permeability is affected mostly by molecular weight and lipophilicity, while additional factors include tissue thickness and melanin content among different species. Loch et al. compared the permeability coefficients of four ophthalmic drugs: lidocaine, ciprofloxacin, timolol, and dexamethasone, through multiple tissue layers in rabbit, porcine, and bovine eyes with Ussing chambers [18], and found that the experimental retina-RPE-choroid-sclera permeability data of porcine ocular tissues were as follows: lidocaine, 8.9 ± 2.1; ciprofloxacin, 1.4 ± 0.2; timolol, 2.4 ± 0.5; dexamethasone, 2.2 ± 0.4; mean ± standard error of the mean, ×10^−6^ cm/s). The smaller molecular weight and lipophilic drug lidocaine (288.8 Da, log *P* = 1.54) revealed the highest permeability, whereas the larger molecular weight and hydrophilic drug ciprofloxacin (367.8 Da, log *P* = −0.54) exhibited the lowest permeability. Recently, Ramsay et al. measured the bovine RPE-choroid permeability of eight small molecular weight drugs and bevacizumab antibody, and found that the outward permeability for ganciclovir was 9.70 ± 7.90, and bevacizumab was 0.0035 ± 0.0020 (mean ± standard error, ×10^−6^ cm/s) [19].

Other than experimental data, some researchers have developed a simulated model to focus on the retinal permeability and clearance mechanism. Hutton-Smith et al. established a three-compartment (aqueous, vitreous, and retina) semi-mechanistic model to interpret the intraocular distribution and elimination process, including the ILM and RPE permeability [40]. In this study, we derived the prediction values of ocular half-life and retinal permeabilities with reference to the intravitreal injection of therapeutic antibodies in rabbit eyes. The authors suggested that the estimated PK parameters were dependent on the molecule sizes presented as hydrodynamic radii, which was consistent with the ex vivo bovine eye measurements. In addition, Haghjou et al. designed a simulated intraocular pharmacokinetics best-fit model to determine the retina-choroid-sclera (RCS) permeability for ophthalmic drugs in the outward direction from the vitreous to systemic circulation [41]. The authors suggested that the prediction of RCS permeability using molecular physicochemical properties might be useful in drug development.

This study had some limitations. First, we should be cautious in extrapolating these data because the estimated neural retina permeability data might not coincide with the real values; because we calculated the neural retina permeability indirectly by the equation with experimental values of retina-RPE-choroid-sclera (RCS) and RPE-choroid-sclera (CS). As the neural retinal layer is very difficult to separate accurately from the RPE-choroid, our method of scraping off the neural retina and measuring the permeability of RCS and CS was an inevitable option. Second, we performed experiments using only three ophthalmic drugs from one animal species. Previous studies have used larger numbers of drug molecules and different species, such as rabbit, porcine, and bovine ocular tissues. Accordingly, we could not derive a statistical correlation between the molecular physicochemical properties and permeability. Moreover, instead of using rabbit eyes, previous studies have analyzed the association between intravitreal half-life and permeability using bovine and porcine eyes to measure the permeability coefficients; thus, direct comparison might be difficult. Finally, since we experimented only in the outward (vitreous to choroid) direction of permeation, there might be a measurement error, even though we assessed the drug concentration after reaching the steady-state time points.

Nonetheless, our study has some strengths that need to be addressed. To the best of our knowledge, this is the first study to analyze and document the neural retinal permeability of ophthalmic drugs. Almost all prior experiments performed permeability studies with RPE-choroid tissues; therefore, in this study, we determined the permeability differences of the neural retina and RPE-choroid tissues. Another strength of our study is that we demonstrated a negative association between intravitreal half-life and permeability in 10 different ophthalmic drug molecules, including the large molecular weight biologic drug bevacizumab, even though the experimental values were obtained for different animal species. To be comprehensive, we will extend the permeability studies with diverse molecules and ex vivo experiments examining both outward and inward directions in bovine, porcine, and rabbit eyes.

In conclusion, we evaluated the RCS and CS permeability of three ophthalmic drugs and calculated the neural retinal permeability. As a result, both molecular weight and lipophilicity are important PK parameters in the posterior segment drug clearance; however, drug transport and permeation in the neural retinal layer are likely to be affected largely by molecular weight. In addition, the intravitreal half-lives of these molecules may be negatively associated with the permeability coefficients. Subsequent studies should be carried out to determine the exact correlation between molecular physicochemical properties, drug permeation, and intraocular pharmacokinetics, represented as intravitreal half-lives.

## Figures and Tables

**Figure 1 pharmaceutics-13-00655-f001:**
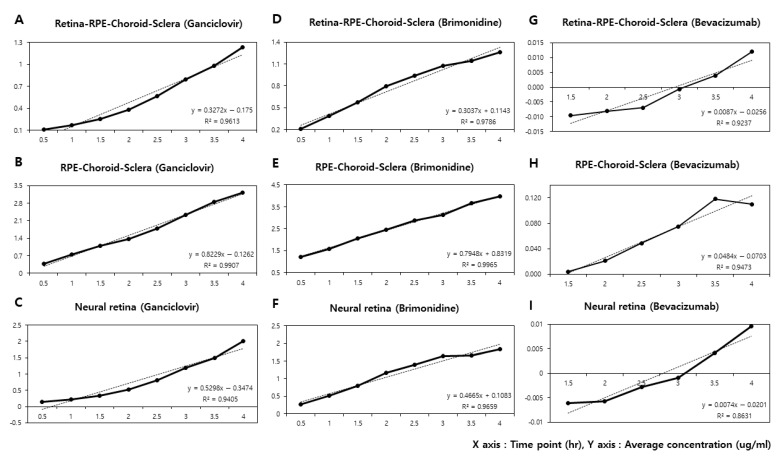
The average time–concentration curves of three ophthalmic drugs in two different rabbit ocular tissues: Retina-PRE-Choroid-Sclera (RCS) and RPE-Choroid-Sclera (CS), and the calculated data of the neural retina. The slope $\frac{\mathrm{dC}}{\mathrm{dt}}$ indicates the change in the acceptor concentration between two time points, and the linear fit models and correlation coefficients (R^2^) are documented. (**A**) RCS, brimonidine: y = 0.3037x + 0.1143 (R^2^ = 0.9786); (**B**) CS, brimonidine: y = 0.7948x + 0.8319 (R^2^ = 0.9965); (**C**) Neural retina, brimonidine: y = 0.4665x + 0.1083 (R^2^ = 0.9659); (**D**) RCS, ganciclovir: y = 0.3272x − 0.175 (R^2^ = 0.9613); (**E**) CS, ganciclovir: y = 0.8229x − 0.1262 (R^2^ = 0.9907); (**F**) Neural retina, ganciclovir: y = 0.5298x − 0.3474 (R^2^ = 0.9405); (**G**) RCS, bevacizumab: y = 0.0087x − 0.0256 (R^2^ = 0.9237); (**H**) CS, bevacizumab : y = 0.0484x − 0.0703 (R^2^ = 0.9473); (**I**) Neural retina, bevacizumab: y = 0.0074x − 0.0201 (R^2^ = 0.8631).

**Figure 2 pharmaceutics-13-00655-f002:**
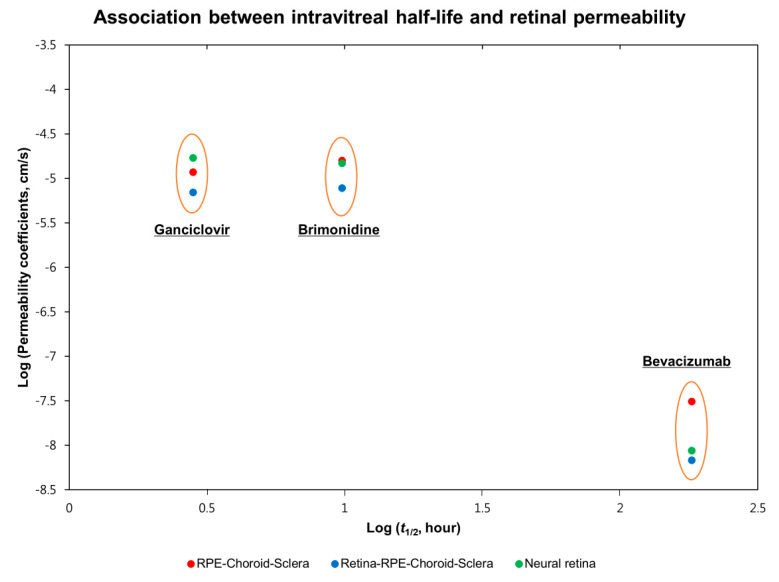
The association between intravitreal half-life and retinal permeability. The experimental data values are documented in Table 2, and then presented as a graph in log scale. Bevacizumab showed low permeability in the ocular tissues due to its high molecular weight compared to brimonidine and ganciclovir.

**Table 1 pharmaceutics-13-00655-t001:** Average time–concentration values and average change of drug concentration at the acceptor chamber.

Time Point (h)	Ganciclovir (*n* = 5)		Brimonidine (*n* = 5)		Bevacizumab (*n* = 5)	
	RCS (μg/mL)	CS (μg/mL)	RCS (μg/mL)	CS (μg/mL)	RCS (μg/mL)	CS (μg/mL)
0.5 h	0.108	0.382	0.215	1.216	−0.009	−0.009
1.0 h	0.172	0.753	0.386	1.578	−0.007	−0.10
1.5 h	0.254	1.084	0.571	2.057	−0.010	0.004
2.0 h	0.383	1.375	0.793	2.465	−0.008	0.021
2.5 h	0.562	1.792	0.937	2.873	−0.007	0.049
3.0 h	0.793	2.341	1.074	3.124	−0.001	0.075
3.5 h	0.981	2.859	1.146	3.681	0.004	0.118
4.0 h	1.237	3.217	1.258	3.967	0.012	0.110
$\frac{\mathrm{dC}}{\mathrm{dt}}$ (μg/mL·h)	0.3272	0.8229	0.3037	0.7948	0.0087	0.0484

RCS: Retina-RPE-Choroid-Sclera; CS: RPE-Choroid-Sclera.

**Table 2 pharmaceutics-13-00655-t002:** Permeability coefficients from the Ussing chamber experiments with rabbit eye tissues.

Parameters	Ganciclovir (*n* = 5)	Brimonidine (*n* = 5)	Bevacizumab (*n* = 5)
Molecular weight (Da)	255.23	292.13	149,000
Log P	−1.66	1.7	-
Water solubility (mg/mL)	4.3	1.5	-
Intravitreal half-life (*t*_1/2_, h)	2.83 ^‡^ [28]	9.9 ^†^ [29]	181.4 ^‡^ [30]
Permeability coefficients (*P*_app_) (×10^−6^ cm/s)			
Retina-RPE-Choroid-Sclera (RCS)	13.78 ± 5.82	15.34 ± 7.64	0.0136 ± 0.0059
RPE-Choroid-Sclera (CS)	23.22 ± 9.74	31.56 ± 12.46	0.0612 ± 0.0264
Neural Retina (R) *	33.89 ± 12.64	29.83 ± 11.58	0.0205 ± 0.0074

Data are expressed as the mean ± standard deviation. ^†^ Dutch belted rabbit; ^‡^ New Zealand White rabbit. * Calculated by the equation of 1/P_RCS_ = 1/P_R_ + 1/P_CS__._

**Table 3 pharmaceutics-13-00655-t003:** Physicochemical properties, intravitreal half-lives, and permeability coefficients of ophthalmic drug molecules in previous literature.

Molecules	Molecular Weight (Da)	Log P	Intravitreal Half-Life (*t*_1/2_, h) *	Permeability Coefficients (*P*_app_) (×10^−6^ cm/s) ^#^	Species	Reference
Aztreonam	434.44	−4.4	7.5 [31]	5.37 ± 5.19	Bovine	Ramsay et al., 2019 [19]
Methotrexate	454.45	−0.241	7.6 [32]	9.39 ± 2.74	Bovine	Ramsay et al., 2019 [19]
Ciprofloxacin	331.3	1.313	4.41 [33]	9.52 ± 5.28	Bovine	Ramsay et al., 2019 [19]
Fluconazole	306.27	0.5	3.18 [34]	15.64 ± 4.66	Bovine	Ramsay et al., 2019 [19]
Voriconazole	349.31	0.927	2.5 [35]	25.00 ± 6.12	Bovine	Ramsay et al., 2019 [19]
Dexamethasone	472	0.65	3.5 [36]	8.90 ± 1.6	Porcine	Loch et al., 2012 [18]
Ketorolac	376.41	2.1	3.09 [37]	69.21 ± 31.9	Bovine	Ramsay et al., 2019 [19]

Data are expressed as the mean ± standard deviation. * Experimental intravitreal half-life data were from New Zealand white rabbit eyes. ^#^ Experimental permeability coefficients were measured as RPE-choroid ocular tissues.

## Data Availability

The datasets used and/or analyzed during the current study are available from the corresponding author on reasonable request.

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
