# Peer review of "Permeability of the Retina and RPE-Choroid-Sclera to Three Ophthalmic Drugs and the Associated Factors"

_pharmaceutics, 2021, doi:10.3390/pharmaceutics13050655_

Round 1
Reviewer 1 Report
This is a well designed study aimed at determination of the apparent permeability coefficients for three drugs, ganciclovir, brimonidine and Bevacizumab across tissues of the retinal blood barrier, specifically the retina/RPE-choroid/sclera and RPE-choroid/sclera.
This work is novel and provides fundamental new knowledge for the field ocular drug delivery. Previous studies have primarily addressed the permeability across either sclera or RPE, but not the tissues described herein. Additional significance is provided by the use of NZW rabbit as a primary species for ocular pharmacology and toxicology studies.
However, there are some concerns that should be addressed before this manuscript is acceptable for publication:
- Table 1 lists "time-concentration values". It would be useful to re-label this to reflect back to the formula (i.e. dC/dT)
- This reviewer has some concerns regarding the linear range used to determine Papp values. For example, for retina-RPE-choroid-sclera with ganciclovir, it appears that it takes up to 2 h to reach study state. Including earlier timepoints will inevitable result in skewed Papp values.
- Previous studies have performed studies in bovine, porcine and other tissues. Given the extensive comparisons performed, it is surprising that the authors did not include a literature-based inter-species comparison - or discuss why this may not be possible.
- The references listed in the Table are not in the bibliography! For example, the Ramsay paper is not referenced and should be: Ramsay E, Hagström M, Vellonen KS, Boman S, Toropainen E, Del Amo EM, Kidron H, Urtti A, Ruponen M. Role of retinal pigment epithelium permeability in drug transfer between posterior eye segment and systemic blood circulation. Eur J Pharm Biopharm. 2019 Oct;143:18-23. doi: 10.1016/j.ejpb.2019.08.008. Epub 2019 Aug 13. PMID: 31419586.
- At times, units do not use the correct Greek symbols, e.g. uM instead of µM. This should be corrected.
Overall, this is an important study characterizing permeability of these important drugs. A revised version is anticipated to make a nice contribution to the Journal.
Author Response
- Table 1 lists "time-concentration values". It would be useful to re-label this to reflect back to the formula (i.e. dC/dT)
à As the reviewer’s suggestion, “Average time-concentration values” in Table 1 was re-labeled to “Average time-concentration values and average change of drug concentration at the acceptor chamber” to reflect the permeability coefficient formula clearly. Moreover, we added the column of dC/dt to reflect the formula easily. In the result section, we re-calculated the data and revised the permeability coefficients in Table 2. Also for the Figure 1, we revised the graphs to fit the re-labeled calculations.
- This reviewer has some concerns regarding the linear range used to determine Papp values. For example, for retina-RPE-choroid-sclera with ganciclovir, it appears that it takes up to 2 h to reach study state. Including earlier timepoints will inevitable result in skewed Papp values.
à Thank you for your suggestion. As we have mentioned in the Methods, we derived the slope of the time-concentration curve between the final time point (4 h) and the initial time point, in our case, 30 min for brimonidine and ganciclovir, and 1.5 hours for bevacizumab. Even though the ganciclovir looks like reaching steady state up to 2 h, comparing to the other 2 molecules, we thought 30 min is the reasonable time point for small molecule such as ganciclovir to calculate the Papp. Since there might be errors in measuring the concentration level at the acceptor chamber, we addressed the limitation of measurement error in the discussion section (Page 15, line 395).
- Previous studies have performed studies in bovine, porcine and other tissues. Given the extensive comparisons performed, it is surprising that the authors did not include a literature-based inter-species comparison - or discuss why this may not be possible.
à In the discussion, Page 15, line 387-394, we addressed the limitations of this study regarding inter-species comparison. “Previous studies have used larger numbers of drug molecules and different species, such as rabbit, porcine, and bovine ocular tissues. Accordingly, we could not derive a statistical correlation between the molecular physicochemical properties and permeability. Moreover, previous studies have analyzed the association between intravitreal half-life and permeability using bovine and porcine eyes to measure the permeability coefficients instead of rabbit eyes; thus, direct comparison might be difficult.” In our study, we used rabbit eye and other literatures have experimented with bovine and porcine tissues. Thus, we thought that the direct comparison might not be possible. In Figure 2 and Table 3, we presented other molecules with different species included in the previous studies (Ramsay et al. and Loch et al.) to show the association between intravitreal half-life and retinal permeability. However, another reviewer pointed out that the experimental data dealing with different species should not be analyzed altogether.
- The references listed in the Table are not in the bibliography! For example, the Ramsay paper is not referenced and should be: Ramsay E, Hagström M, Vellonen KS, Boman S, Toropainen E, Del Amo EM, Kidron H, Urtti A, Ruponen M. Role of retinal pigment epithelium permeability in drug transfer between posterior eye segment and systemic blood circulation. Eur J Pharm Biopharm. 2019 Oct;143:18-23. doi: 10.1016/j.ejpb.2019.08.008. Epub 2019 Aug 13. PMID: 31419586.
à Thank you for your comment. It was our mistake of citing errors probably due to Endnote program. As the reviewer’s indication, we modified the references in Table 3.
- At times, units do not use the correct Greek symbols, e.g. uM instead of µM. This should be corrected.
à As reviewer’s indication, we corrected wrong Greek symbols in the manuscript, Tables, and Figures.

Reviewer 2 Report
The manuscript is focused on ocular drug permeability studies, using an ex vivo rabbit eye tissue model. The results presented in the manuscript are interesting, confirming previously published data, highlighting the importance of some pharmacokinetic parameters for a more efficient ophthalmic drug delivery. However, several issues must be corrected, or better explained in the manuscript:
1. The authors should specify in the manuscript how they monitored the integrity of the tissues during the experiments.
2. The authors should modify Figure 2, because the presented data came from two different experiments (their own data and those from Ramsay et al, 2019) which used tissues from different species (rabbit vs. bovine) with different characteristics. It would be better for the authors to present only their data, despite the fact there are only three drugs, still the differences between the three are relevant.
3. Inconsistencies and self-contradictory affirmations in the manuscript should be corrected, for example in the caption of Figure 2, the authors state that "longer intravitreal half-life drugs tend to show greater permeability in the ocular tissues, especially bevacizumab (Lines 250-251). It is known that bevacizumab, being a protein, has a poor permeability like the authors stated earlier in the manuscript (Line 213).
4. There are a lot of citing errors in the text, for example in Table 2 for T1/2 of bevacizumab, the presented reference (28) is about aztreonam. Furthermore, in Table 3, you quoted Ramsay et al. as being reference no. 34, when it is actually no. 19. Please read the text carefully and correct all the errors regarding references.
Author Response
Reviewer #2
The manuscript is focused on ocular drug permeability studies, using an ex vivo rabbit eye tissue model. The results presented in the manuscript are interesting, confirming previously published data, highlighting the importance of some pharmacokinetic parameters for a more efficient ophthalmic drug delivery. However, several issues must be corrected, or better explained in the manuscript:
- The authors should specify in the manuscript how they monitored the integrity of the tissues during the experiments.
à To preserve the integrity of ocular tissues and maintain a constant pH during permeation experiments, we used ocular tissues immediately after enucleation of rabbit eyes and stored in ice-cold Ringer buffer or PBS (pH 7.4). Also, PBS buffer (pH 7.4) was used as donor and receiving solution, and the Ussing chamber was filled with oxygen or a mixture of 95% O2 and 5% CO2 in both compartments. Though we didn’t do histological analysis for ocular tissues, we guess that immediate usage of ocular tissues and proper buffer such as Ringer solution can minimize the damage to the tissue integrity. As reviewer’s comment, the sentences “To minimize the damage of tissue integrity, the extracted tissues were used immediately” and “Both compartments in the Ussing chamber was filled with oxygen or a mixture of 95% O2 and 5% CO2” were added. (Lines 129-130 and 141-142)
- The authors should modify Figure 2, because the presented data came from two different experiments (their own data and those from Ramsay et al, 2019) which used tissues from different species (rabbit vs. bovine) with different characteristics. It would be better for the authors to present only their data, despite the fact there are only three drugs, still the differences between the three are relevant.
à Thank you for your suggestion. At first, we wanted to gather other molecular data from Ramsay et al. and Loch et al. to present the association between intravitreal half-life and retinal permeability. However, as the reviewer has pointed out, analyzing data from different species with different characteristics are not plausible. Therefore, we modified Figure 2 with presenting only our data, despite the fact there are only three drugs.
à Also, we erased the Spearman correlation coefficient, since then we could not derive the statistical result with only 3 molecules, and modified the last paragraph in the Result section: “In addition, we evaluated the association between intravitreal half-life and posterior segment ocular tissue permeability with the experimental data of three ophthalmic drugs from this study. The log scale graph chart is shown in Figure 2, and we propose that the large molecule of bevacizumab with longer intravitreal half-life showed smaller permeability coefficient than brimonidine and ganciclovir.”
- Inconsistencies and self-contradictory affirmations in the manuscript should be corrected, for example in the caption of Figure 2, the authors state that "longer intravitreal half-life drugs tend to show greater permeability in the ocular tissues, especially bevacizumab (Lines 250-251). It is known that bevacizumab, being a protein, has a poor permeability like the authors stated earlier in the manuscript (Line 213).
à Thank you for your comment. It was our mistake of self-contradictory affirmations in the manuscript. As the reviewer’s suggestion, the sentence “longer intravitreal half-life drugs tend to show greater permeability in the ocular tissues, especially bevacizumab” (Lines 296-298) was modified to the sentence “Bevacizumab showed low permeability in the ocular tissues due to its high molecular weight compared to brimonidine and ganciclovir (line 295-296).”
- There are a lot of citing errors in the text, for example in Table 2 for T1/2 of bevacizumab, the presented reference (28) is about aztreonam. Furthermore, in Table 3, you quoted Ramsay et al. as being reference no. 34, when it is actually no. 19. Please read the text carefully and correct all the errors regarding references.
à Thank you for your suggestion. It was our mistake of citing errors probably due to Endnote program. We corrected the references in Table 2.

Round 2
Reviewer 2 Report
I am glad that the authors have corrected the manuscript, according to suggestions, therefore I recommend the publication of their article.